# The Real-World Evidence on the Fragility and Its Impact on the Choice of Treatment Regimen in Newly Diagnosed Patients with Multiple Myeloma over 75 Years of Age

**DOI:** 10.3390/cancers15133469

**Published:** 2023-07-02

**Authors:** Agata Tyczyńska, Marcela Krzysława Krzempek, Alexander Jorge Cortez, Artur Jurczyszyn, Katarzyna Godlewska, Hanna Ciepłuch, Edyta Subocz, Janusz Hałka, Anna Kulikowska de Nałęcz, Anna Wiśniewska, Alina Świderska, Anna Waszczuk-Gajda, Joanna Drozd-Sokołowska, Renata Guzicka-Kazimierczak, Kamil Wiśniewski, Agnieszka Porowska, Wanda Knopińska-Posłuszny, Janusz Kłoczko, Piotr Rzepecki, Dariusz Woszczyk, Hanna Symonowicz, Grzegorz Władysław Basak, Barbara Zdziarska, Krzysztof Jamroziak, Jan M. Zaucha

**Affiliations:** 1Department of Hematology and Transplantology, Medical University of Gdańsk, 80-214 Gdańsk, Poland; atyczynska@uck.gda.pl; 2Department of Biostatistics and Bioinformatics, Maria Sklodowska-Curie National Research Institute of Oncology, Gliwice Branch, 44-102 Gliwice, Poland; marcela.krzempek@gliwice.nio.gov.pl (M.K.K.); alexander.cortez@gliwice.nio.gov.pl (A.J.C.); 3Plasma Cell Dyscrasia Center, Department of Hematology, Jagiellonian University Medical College, 31-008 Kraków, Poland; 4Department of Hematology, Medical University of Bialystok, M. Sklodowskiej-Curie 24A, 15-276 Bialystok, Poland; 5Copernicus Regional Oncology Center, 80-803 Gdansk, Poland; 6Department of Hematology, Military Institute of Medicine, 01-755 Warsaw, Poland; 7Department of Hematology, State Hospital, 45-221 Opole, Poland; 8Department of Oncology and Chemotherapy, Nicolas Copernicus State Hospital, 75-581 Koszalin, Poland; 9Department of Hematology, University Hospital of Karola Marcinkowski in Zielona Góra, 65-046 Zielona Góra, Poland; 10Department of Hematology, Transplantation and Internal Medicine, Medical University of Warsaw, 02-091 Warsaw, Poland; 11Department of Hematology, Pomeranian Medical University, 70-204 Szczecin, Poland; 12Department of Hematology, Institute of Hematology and Transfusion Medicine, 02-776 Warsaw, Poland; 13Department of Oncology and Hematology, Central Clinical Hospital, Ministry of the Interior, 01-150 Warsaw, Poland; 14Department of Hematology, Gdynia Oncology Center, 81-519 Gdynia, Poland

**Keywords:** frailty, elderly, multiple myeloma, choice of treatment, over 75 years old, early mortality

## Abstract

**Simple Summary:**

About 35% of patients with multiple myeloma are ≥75 years old at diagnosis, and their treatment should be individualized according to their fragility score. Fragility assessment scales include the International Myeloma Working Group Palumbo Fragility Scale, the Revised Initial Myeloma Comorbidity Index, and the Mayo score. However, they are time-consuming, but more importantly, there are no convincing data from a real world experience supporting their usefulness in everyday clinical practice. Patients with multiple myeloma over 75 years of age are instantly classified as intermediate fit or frail. These patients should be offered non-intensive treatment in reduced doses or palliative care. In the prospective observational, multicenter study, we showed that most patients over 75 years of age qualified for treatment are safely treated with three-drug regimens regardless of their fragility categorization. Current fragility scores for patients with MM over 75 years old have limited value in daily practice.

**Abstract:**

Fragility scales are intended to help in therapeutic decisions. Here, we asked if the fragility assessment in MM patients ≥ 75 years old qualified for treatment by the local physician correlates with the choice of treatment: a two- or three-drug regimens. Between 7/2018 and 12/2019, we prospectively enrolled 197 MM patients at the start of treatment from the 13 Polish Myeloma Group centers. The data to assess fragility were prospectively collected, but centrally assessed fragility was not disclosed to the local center. The activity of daily living (ADL) could be assessed in 192 (97.5%) and was independent in 158 (80.2%), moderately impaired in 23 (11.7%), and 11 (5.6%) in completely dependent. Patients with more than three comorbidities made up 26.9% (53 patients). Thus, according to the Palumbo calculator, 43 patients were in the intermediate fitness group (21.8%), and the rest belonged to the frailty group (153, 77.7%). Overall, 79.7% of patients (157) received three-drug regimens and 20.3% (40) received two-drug regimens. In each ECOG group, more than three out of four patients received three-drug regimens. According to the ADL scale, 82.3% of the independent 65.2% of moderately impaired, and 81.8% of the dependent received three-drug regimens. Out of 53 patients with at least four comorbidities, 71.7% received three-drug regimens, and the rest received two-drug regimens. Thirty-four patients from the intermediate fit group (79.0%), and 123 (79.9%) from the frail group received three-drug regimens. Early mortality occurred in 25 patients (12.7%). No one discontinued treatment due to toxicity. To conclude, MM patients over 75 are mainly treated with triple-drug regimens, not only in reduced doses, regardless of their frailty scores. However, the absence of prospective fragility assessment did not negatively affect early mortality and the number of treatment discontinuations, which brings into question the clinical utility of current fragility scales in everyday practice.

## 1. Introduction

It is estimated that the percentage of people over 75 years of age will increase up to 22% in developed countries by 2023, which means that, as described in 1976 by Isaac Bernard, “great geriatric problems” experienced by frail people will impact the treatment of cancer [1]. Fragility is a multidimensional state of reduced reserves of energy, fitness, cognitive ability, and health that makes individuals more vulnerable to stressors and can reduce resistance to cancer and cancer treatment. Understanding areas of vulnerability, such as frailty, comorbidities, and disability, is believed to be important for tailoring treatment to optimize outcomes for individual patients [2,3,4,5].

The Polish Gerontological Society recommends that individuals over 60 years old should undergo a comprehensive geriatric assessment. The comprehensive geriatric assessment is a multi-directional diagnostic process for obtaining data on a patient’s medical, psychosocial, and functional well-being. It includes functional fitness, mental health, socio-environmental conditions, and physical health. This approach is achieved through valorized research in the form of questionnaires and tests that take approximately 20 to 40 min to complete. It helps understand the patient and their needs and indicates the possibilities for social functioning and therapeutic disease processes. Moreover, it enables identifying patients who, despite their age, can receive full oncological treatment without an increased risk of complications and those who only qualify for supportive and symptomatic treatment [4,6]. The intermediate group consists of patients requiring therapy modification to reduce side effects and maintain quality of life.

Most patients with multiple myeloma (MM) are over 65 years of age (60% of patients). Modern treatment of MM contains two to four agents: steroids with either alkylating agents or proteasome inhibitors, or/and immunomodulatory agents, and recently monoclonal antibodies. Multiple agent schemas are usually more effective but also more toxic. This is particularly important for elderly patients who are more sensitive to complications as they have smaller organ reserves, less ability to repair cellular damage, comorbidities, polypharmacy, and slower pharmacokinetics and pharmacodynamics of drugs. Using scales to assess their fragility to optimize therapy is reasonable. Frailty assessment of the elderly was initially based on age and performance status, according to Karnofsky and the Eastern Cooperative Oncology Group (ECOG). However, this proved to be insufficient over time. The first scale proposed by the International Myeloma Working Group (IWMG) to assess myeloma patients from three different groups was the Geriatric Assessment scale (GA) introduced by Palumbo et al., which has been validated based on four prospective studies of 869 patients [4,7]. The GA scale includes the activities of daily living (ADL, Katz scale), instrumental ADL (I-ADL, Lawton scale), and the Charlson Comorbidity Index (CCI), in addition to age and ECOG performance status, and distinguishes three groups of patients: fit—*fit*, intermediately fit—*intermediate fit*, and weak (frail)—*frail*. Of these groups, the *frail* group is associated with a more than three-fold higher risk of death, progression, and adverse events, regardless of International Staging System (ISS) prognosis, genetic disorders, and type of treatment (Appendix A).

The authors suggest that patients from the *fit* group could receive the full dose of recommended treatments and may benefit from autologous transplantation; patients from the intermediate group may benefit from treatment with modified doses or a two-drug treatment (Appendix A). In contrast, *frail* patients may benefit from palliative, symptomatic, or reduced treatment doses after carefully analyzing possible side effects [4] (Appendix A). An important observation in developing the GA fitness calculator is that 80 years of age should be the limit for defining a “weak state” [4,7].

The second scale evaluating the efficiency of patients with MM is the Revised Initial Myeloma Comorbidity Index (R-MCI), which was validated by a retrospective evaluation of 801 patients (1997–2012) from a center in Germany [8]. The scale uses five points concerning the patient’s condition to determine a score, which includes kidney disease, lung disease, Karnofsky’s fitness rating scale, age, and fitness [9]. Patients were divided into three groups with a clear relationship between fitness and treatment tolerance and significantly different overall survival (OS) times. In the *fit* group, OS was 10.1 years, while it was 4.4 years in the intermediately *fit* group and 1.2 years in the *frail* (weak) group. Lung and kidney diseases significantly affect treatment tolerance regardless of the treatment used. However, the greatest limitation of the R-MCI scale is that the therapies used in this study are currently not recommended (Appendix A).

The third proposed prognostic scale used for people over 65 is the Mayo Risk Score (MRS) [10], assessing the ADL, CCI, ECOG performance status, and Revised-ISS (R-ISS) stage. In addition, N-terminal pro-brain natriuretic peptide (NT-proBNP) protein levels were measured. The study confirmed that three parameters are sufficient to identify the “weak” group for whom treatment may be detrimental and shorten overall survival (OS). The components of the scale are age, ECOG, and N-terminal pro B-type natriuretic peptide (NT-proBNP). NT-proBNP indicates cardiac and renal function and a threshold of 300 ng/L corresponds to a well-established age-independent cut-off point for excluding acute heart failure [11]. Patients over 70 years of age with ECOG performance ≥ 2 and NT-proBNP ≥ 300 ng/L belonged to the high group, and their OS time was 18 months, with only 18% of them receiving any therapy. The second group of patients met two out of three criteria and had an OS of 28 months, while OS was 58 months in patients with one point [10].

The presented fragility assessment scales are valuable, but are challenging to use in everyday clinical practice. Indeed, the scale proposed by the IMWG is time-consuming, taking around 40 min to complete. The R-MCI scale seems simpler and less time-consuming. The patient completes the test themselves, which may therefore be biased. The MRS, based on three parameters, is the most straightforward tool in everyday practice. However, the lack of an objective assessment of the patient’s functioning in everyday life is a limitation of this tool. Patients over 75 are automatically classified as immediately fit or frail according to the IMWG scale. According to the guidelines, these patients should be offered non-intensive treatment at reduced doses. Palliative care and supportive treatment are indicated in the fragile group, and treatment should include reduced doses of at most two-drug regimens [11] (Appendix A).

Despite the above recommendation, no data from real-world evidence (RWE) studies support the clinical utility of current fragility scales in everyday practice. Therefore, we asked if the subjective clinical assessment of the local treating physician and the subsequent choice of the intensity and type of treatment: two- versus three-agent schema, corresponds to the prospective assessment of the fragility score in MM patients over 75 years. We also assess whether more intensive treatment was associated with worse tolerance and first-line mortality.

## 2. Materials and Methods

### 2.1. Study Participants

The prospective, cross-sectional observational study performed between July 2018 and December 2019 enrolled 197 MM patients ≥ 75 years at the start of treatment from the 13 Polish Myeloma Group centers. The palliative patients were not included. The patients were followed for a fixed 12-month period.

### 2.2. The Fragility Assessment

The centers were asked to collect the following data at the time of diagnosis: clinical stage of the disease according to the ISS and Durie and Salmon classification, CCI, performance status according to the ECOG scale, ADL (Katz scale) and I-ADL (Lawton scale), treatment method, and primary blood test results, including lactate dehydrogenase (LDH), C-reactive protein (CRP), hemoglobin, platelets, NT-pro-BNP, ẞ2-microglobulin, albumin, calcium, and creatinine. The frailty scores were assessed centrally and were not disclosed to the local centers. Primary physicians did not perform the frailty scores assessment. In the second part of the study, data on patient management, assessments of the response to treatment, survival, and post-treatment complications were collected 12 months after the initiation of treatment.

### 2.3. Statistical Analysis

Categorical variables were summarized as frequencies and percentages. Pairwise comparisons between patient subgroups employed Fisher’s exact test. The existence of a monotonic trend was assessed with the Cochran–Armitage test. The correlation of treatment regimen with frailty scores was investigated using the rcompanion (v. 2.4.30) and psych (v. 2.3.6) packages (Cramer’s V or Phi correlation was applied accordingly). All the analyses were performed using the R software package version 4.0.1 (R Foundation for Statistical Computing, http://www.r-project.org (accessed on 15 June 2020)). A two-sided *p*-value ≤ 0.05 was considered statistically significant. All data were entered into a Microsoft Office Excel spreadsheet.

## 3. Results

### 3.1. Patients Characteristics

The cohort included 100 males and 97 females at a median age 77 (75–90) years. One third of patients were over the age of 80 (65, 33.0%). The clinical characteristics are shown in Table 1. Most patients were diagnosed with myeloma of the immunoglobulin (Ig)G subtype (69.0%), followed by IgA (16.2%) and light chains (10.2%), IgM (2.0%), IgD (1.0%), and biclonal type (0.5%).

### 3.2. Physical Fitness and Comorbidities

ECOG 0-2 was documented in 171 (86.8%) patients (Table 2. The simple geriatric scale of basic vital functions according to the ADL was assessed in 192 (97.5%) patients, and most of them (82.3%) were independent (Table 2). All patients were evaluated with the CCI, which revealed that only 53 (26.9%) had more than three comorbidities. The most common comorbidities were hypertension (160, 81.2%), ischemic heart disease (58, 29.4%), heart failure (50, 25.4%), kidney disease (48, 24.4%), respiratory disease (41, 20.8%), diabetes (40, 20.3%), gastritis/gastroesophageal reflux disease (29, 14.7%), other cancers (28, 14.2%), hepatic dysfunction (22, 11.2%), and cerebral circulation disorders (18, 9.1%). Less frequent comorbidities included dementia (6, 3.0%), connective tissue disease (4, 2.0%), and hemiparesis (4, 2.0%).

### 3.3. Evaluation of the First-Line Treatment

A three-drug regimen was used in 157 (79.7%) of patients. The most frequent schemas were CTD 54 (27.4%) and VMP 39 (19.8%), then MPT, VTD and VCD: 31 (15.7%), 22 (11.2%), 10 (5.1%), respectively (Table 3). There was no preference for any two-drug regimen (Table 3).

Modification of the scheme consisted of reducing the dose of at least one drug. Among the three-drug regimens, 22.3% (35 patients) were modified, as were 47.5% (19 patients) of the two-drug regimens. The most frequently modified regimen was the VTD (13 out of 22) and VCD (3 out of 10) regimens, while among the two-drug regimens, the modification was similar in percentage in each regimen.

Of the patients, intermediate fit (43; 21.8%) and frailty (154; 78.2%) according to the IMWG scale, the modification (reduction of doses) in three-component therapies was 27.2% for intermediate fit, and 22.8% for frailty. It is noteworthy that 47.5% of patients treated with two-component regimens (19 of 40; 20.3%) had reduced dose regimens, and similarly in both groups: intermediate fit 44.4%, and fragile 49.4% (Appendix A).

It is worth noting that among the patients over 80 years of age (65 patients, 33.0%), 50 (76.9%) also received the three-drug therapy (and 30% of them received a modified regimen).

No case of treatment discontinuation due to unacceptable toxicity or due to patient withdrawal from therapy was reported.

### 3.4. First-Line Treatment Analysis

During the 12 months of prospective observation, 107 (54.3%) patients received one line of treatment, 84 (42.6%) two lines of treatment, and 6 (3.0%) three lines of treatment. Next, we analyzed whether the functioning status and comorbidities correlated with the type of first-line therapy (Appendix A). Over three-quarters of patients in each ECOG fitness group received a three-drug regimen (Table 2). No relationship was found between the general performance status and the type of regimens (two- vs. three-drugs) (Table 2). However, higher creatinine levels (above 1.9 mg/dL) determined the choice of treatment: individuals with elevated creatinine levels were more likely to receive two-drug regimens (OR = 3.05, 95%CI [1.33, 6.92]; *p* = 0.005) (Table 2).

Regarding performance status, according to the ADL, 82.3% of the fit patients and, surprisingly, 81.8% of the completely dependent ones received three-drug regimens (Table 2). The I-ADL could be recorded on the day of enrollment, and after 12 months, for only 59 patients (29.9%). In almost all patients (58, 98.3%), the deterioration of complex activities of daily living was documented, of whom 52 (88.1%) received a three-drug regimen. The correlation between the choice of treatment regimen and the frailty scale (IMWG) was not statistically significant (Table 2).

We also asked if the number of comorbidities affected the choice of treatment. The median number of comorbidities was 2 (0–9). Every fourth person had at least four comorbidities, and 71.7% of the patients in this group received a three-drug treatment. The percentages of patients with a lower number of comorbidities receiving three-drug schemas were similar and comparable to patients with a higher number of comorbidities (Figure 1). The Cochran–Armitage test showed no significant trend (*p* = 0.335) (Figure 1) between the number of comorbidities and the applied treatment regimen (two-drug or three-drug regimen) and additionally, Fisher’s exact test showed no relationship between these variables (*p* = 0.395) (Appendix A).

No statistically significant relationship was found when analyzing the relationship between the type of comorbidity and the selected regimens (Appendix A). 

### 3.5. Evaluation of the Response to First-Line Treatment

Almost three-quarters (143, 72.6%) of patients responded to the first line treatment regardless of the two- or three-drug regimen. Three-drug regimens allowed for a better response (CR + VGPR + PR vs. SD + PD: OR = 2.09, 95% CI [0.93, 4.58]; *p* = 0.072) since patients who received a two-drug regimen achieved more often only disease stabilization (SD vs. rest: OR = 3.304, 95% CI [1.37, 7.76]; *p* = 0.005) (Table 4).

Bortezomib used in the first-line treatment in 83 patients (42.1%) did not bring any benefit in terms of better responses (*n* = 197, *p* = 0.11) (Appendix A).

### 3.6. Evaluation of the Response Performance Status and Fragility

The response to treatment after first-line treatment was analyzed according to ECOG performance status, ADL, and the presence of the most common comorbidities (Table 5). None except one of the 26 patients with ECOG > 2 achieved CR or VGPR. Similar trend was observed for the ADL (Table 5). None of the 11 completely dependent patients achieved CR or VGPR. Moreover, many of them (36.4%) experienced disease progression, and only 45.5% responded to treatment, regardless of the treatment used (Table 5). The response rates obtained in patients with at least four comorbidities were no worse than in the other groups, and there was no relationship between the number of comorbidities and the response (*p* = 0.156).

### 3.7. Analysis of Deaths

During the first-line treatment, fourteen patients died, with six (42.8%) due to disease progression, five (35.7%) to infection, one (7.1%) to heart failure, and one (7.1%) due to other malignancy, while the cause of death was not established in one patient (7.1%) (Appendix A). Of the 14 deceased persons, 12 (85.7%) received a three-drug treatment (four patients with modified treatment, 33.3%), which is 7.6% of all the patients receiving three-drug regimens, and the remaining two had a two-drug regimens (one patient was treated by modified regimen) (Table 2). During the second line of treatment, eleven people died: nine due to disease progression and two in the course of an infection. At the end of the study, 25 (12.7%) patients died in total, 15 (7.6%) due to disease progression and 7 (3.6%) due to infection (Appendix A), after 12 months from the diagnosis of the disease.

## 4. Discussion

This study suggests that the available frailty scales for patients with MM over 75 who were qualified for treatment are not useful in clinical practice regarding the choice of two- vs. three-drug regimens. The early mortality and the number of treatment discontinuations were not affected by the absence of a prospective fragility assessment by the local physician. This surprising observation may result from the fact that our patients, who qualified for treatment despite advanced age (>75 years old), were relatively fit, with the majority being in ECOG 0-2. However, age itself plays a very important role in the fragility assessment. According to the recommended frailty scale developed by the International Myeloma Working Group [7,9,11,12], patients over 75 years can only be categorized as intermediate fit or fragile depending on the number of comorbidities (two or more than two) and patients over 80 years are always categorized as frail. Consequently, all patients from our study should be treated either with a modified drug doses regimen (intermediate fit) or should be referred to palliative treatment or at most, reduced dose two-drug regimens (fragile) [7,11,12,13,14].

Our data showed that even very old frail MM patients might benefit from the three-drug regimens administered by an experienced primary oncologist or hematologist. Nevertheless, the management of these patients still remains a challenge, primarily be-cause there are no clear recommendations on how to treat patients over 75 years of age. So far, there have been no prospective clinical trials dedicated to this group of patients. In addition, in many studies for patients with newly diagnosed myeloma, age over 80 years is an exclusion criterion. Based on the presented RWD, neither age nor the number and/or type of comorbidities (except for renal impairment) affect the choice of treatment and the start of therapy. These observations suggest that age and the number of comorbidities should not be the major determinants of frailty.

Our results are in line with other current RWE that showed that prior to the daratumumab era, most of the MM patients older than >75 were treated with three-drug regimens, most commonly with steroids (dexamethasone at a dose < 20 mg/dose), immunomodulatory drugs (lenalidomide or thalidomide—at the best tolerated dose, starting treatment with the full recommended doses) and cyclophosphamide or bortezomib (both at the best-tolerated dose, starting treatment with the recommended doses of cyclophosphamide and bortezomib of one per week at full dose). The advantage of the CTD and MPT regimen is their oral administration. Therefore, it might be a preferred choice for the physician and the patient and their family/carers alike. We showed that the use of bortezomib did not improve the response, but its use requires more frequent visits to treatment centers, which may pose a logistic problem for elderly patients. 

Our cohort included 22% intermediate-fit patients and 78% frail patients. None of them underwent autotransplantation, but three-drug regimens (also VTD and VCD) were used in about 80% of patients in each group. It should be noted that the modification of drug doses in the 3-component regimens affected every 4th patient in the intermediate-fit group and every 5th patient in the frail group (Appendix A). There-fore, our group of patients was treated more intensively than the accepted recommendations, and the choice was not influenced by objective scales or other parameters. In our study, regimens were used according to the recommendations of the IMWG and the Polish Myeloma Group [12]. However, only 22.3% of patients treated with the three drug regimens required drug modification. The most often modified regimens included VTD, VCD, and CTD. This translated to a better tolerance, although at the expense of a worse treatment response. We demonstrated that the safest three-component regimens are VCD, MPT and CTD, and the least safe are VMP, followed by VTD (based on the number of fatal events in the first and second line of treatment) (Table 3). Notably, VRD was used in only one patient since this schema was not reimbursed at the time of the study. 

It is worth noting that patients with impaired renal function or kidney disease were referred for treatment with a three-drug regimen less frequently and were treated with regimens containing bortezomib more often (50%). Renal impairment usually indicates more aggressive and advanced diseases not always well-reflected by the ISS score [15]. Therefore two drug regimens, especially daratumumab with dexamethasone, might be insufficient for some patients on dialysis, patients with advanced kidney disease, or those with advanced vascular thromboembolism. The results of the NCT04151667 study will probably provide the answer to this question.

In our analysis, during 12 months of treatment, no patient discontinued treatment due to toxicity or self-decision. It is much lower compared to the Palumbo study where the cumulative incidence of treatment discontinuation at 12 months was 20.8% in the intermediate fitness group and 31.2% in the frail patient cohort [4]. If death due to infection or exacerbation of comorbidities can be perceived as the toxicity leading to treatment discontinuation, one should add ten patients (5.1%) who died, of which nine received treatment with three drugs (4.6%) and eight were in the frailty group. This observation indicates that three-drug regimens should be used with caution in frail patients. 

In contrast, our data suggest that three-drug regimens can be safely administered: early mortality (12.7%) was much lower than the 35.3% reported from NCI SEER data from 1993–2010 [16,17] and comparable to the results from newer studies: 9.1% in the study by McQuilten and colleagues [18], and 12.6% in an observational study of the Asian population [19] with the exception of a study by Lee et al., who reported 30% mortality in this group of patients [20]. Additionally, it was shown in our analysis that mortality was similar in both groups, regardless of the treatment intensity (for the three-drug regimen as the first line therapy it was: 7.6%, and for the two-drug regimen it was 5.0%). 

Existing data suggest that mortality during the first-line treatment is mainly related to the exacerbation of comorbidities [21]. In the presented analysis, mortality from exacerbation of comorbidities during therapy occurred only in two patients; both received three-component treatment (1%) (heart failure and lung cancer progression) (Appendix A). This may suggest some selection bias for “better” patients referred to our centers. The high number of patients with ECOG 0-2 corroborates this notion.

The lack of a clear relationship between the current fragility assessment in MM patients may also result from the fact that new drugs registered during the last five years have a significantly different toxicity profile and its effect on older and frail patients. In the MAIA study, in which 44% of the patients over 75 (9% intermediate fit, 32.9% frail), daratumumab (anti-CD38 antibody) along with Rd (Dara-RD) not only extended the median PFS from 34.4 months to 61.9 months and the OS (under development), but also showed good tolerance [22]. Early mortality (up to 12 months) in the MAIA study was 5.8% of patients in the frailty group in the D-Rd arm and 5.3% in the Rd arm, while in the intermediate fit group in the D-Rd arm it was 0.8%, and 2.5% in the Rd arm. On the other hand, the reduction of doses in the frail group in the study arm (Dara-Rd) affected 64.3% of cases, and 44% in the control arm (Rd). Unfortunately, the available data lack information on the subgroup over 75 years of age. Moreover, the complications of this treatment were no greater than those in the control group. Of note, all patients enrolled to the MAIA trial were in ECOG 0-2. This observation, similarly to our findings, suggests that patients with MM over 75 with good performance status (ECOG 0-2) could be safely treated with the three-drug regimen, even though in our study, daratumumab was not used as the first line. However, an additional fourth agent, such as in the ALCYONE study (daratumumab + VMP), resulted in higher toxicity, mainly in terms of infection, and a shorter PFS of 36.4 months [23]. Therefore, four-drug regimens may be too toxic in this group of patients. Currently, the Dara-Rd regimen is the most effective treatment and has an acceptable safety profile. The results of a trial of daratumumab with VRd vs. VRd (NCT03652064) for transplant ineligible patients are eagerly expected with the hope they would answer questions regarding the safety and efficacy of a four-drug regimen in elderly patients. 

New treatment options such as chimeric antigen receptor (CAR)-T therapies and bispecific antibodies (T-cell redirecting therapies) need careful evaluation in patients over 75 years old. The CARTITUDE 5 trial using ciltacabtagene autoleucel (ciltacel) includes patients not eligible for transplantation, but it precludes patients over 80 years old (NCT04923893). CAR-T therapy is well tolerated, and toxicity appears predictable and manageable. The safety data are so far optimistic and demonstrate acceptable rates of Grade 3 and 4 events [24,25], which suggests that this therapy might be also considered in elderly patients, without any age limit, after prior careful selection in order to prevent possible adverse events. In contrast to CAR-T, bispecific antibodies are available “off the shelf” and are most often administered cyclically subcutaneously (every 1–4 weeks). The recruitment to trials for non-transplant eligible patients in initial line of treatment, with no age limit, has recently started (EudraCT Number 2021-000803-20). The safety of bispecific antibodies is even better than CAR-T and patients over 75 could benefit from this therapy. The question of fragility should be addressed again in the era of new therapies.

## 5. Conclusions

To conclude, our study showed that in everyday practice, the management of patients over 75 who qualified for treatment was not dependent on the performance of geriatric scales and the patient’s fragility assessment. The choice of the type of therapy did not correlate with the fragility score. However, most patients had a relatively good performance status (ECOG 0-2). They were treated with three-drug regimens regardless of the number of comorbidities. Despite that, mortality in the first line was acceptable and comparable to other studies. Impaired renal function was the only factor leading to a more frequent choice of two-drug regimens. Our results therefore question the need for a formal geriatric assessment used in its present form in daily practice and suggest that a holistic clinical assessment by an experienced physician is no worse for the appropriate choice of treatment, especially in patients with good performance status. However, a new era in multiple myeloma treatment may require us to re-address the utility of the frailty scores. Future frailty scores should be based on more objective data and not on “physician feeling”, especially in view of the potential severe toxicities of new treatment options (immunotherapy and CAR T cells). A big limitation of our study is the lack of cytogenetic results and its possible effect on the choice of therapy. Moreover, our study did not include a questionnaire assessing the quality of life, which could help to better understand the patient’s views on the chosen treatment.

## Figures and Tables

**Figure 1 cancers-15-03469-f001:**
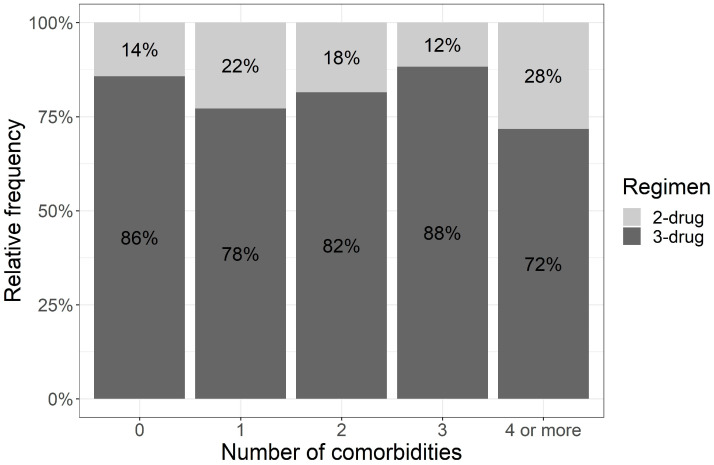
Trend between the number of comorbidities and the treatment protocol selected (test Cochran–Armitage).

**Table 1 cancers-15-03469-t001:** Clinical characteristics.

Characteristic	Number of Cases N = 197 n (%)
Sex: F/M	97 (49.2%)/100 (50.8%)
IgG	136 (69.0%)
IgA	32 (16.2%)
IgD	2 (1.0%)
FLC (light chain)	20 (10.2%)
IgM	4 (2.0%)
LDH activity above the upper normal range	44 (22.3%)
ISS: I	25 (12.7%)
ISS: II	66 (33.5%)
ISS: III	96 (48.7%)
Bone disease Yes/No *	154 (78.2%)/36(18.3%)
Hemoglobin less than 10 mg/dL: Yes/No	101 (51.3%)/96 (48.7%)
Calcium level above 2.75 mmol/L: Yes/No	28 (14.2%)/168 (85.3%)
Creatinine above 1.9 mg/dL: Yes/No	44 (22.3%)/153 (77.7%)

* missing data have been omitted.

**Table 2 cancers-15-03469-t002:** Assessment of the type of treatment and selected parameters.

Characteristic	Number of Cases (*n*)	First-Line Treatment Used (%)		Independence Assessment *p* Value	Correlation Assessment φc or rφ *
		Three-Drug Treatment	Two-Drug Treatment		
**Sex**					
Male	100	80 (80.0%)	20 (20.0%)	0.999	-
Female	97	77 (79.4%)	20 (20.6%)		
**ISS**					**φc**
ISS I	25	21 (84.0%)	4 (16.0%)	0.602	0.081
ISS II	66	55 (83.3%)	11 (16.7%)		
ISS III	96	74 (77.0%)	22 (23.0%)		
**Presence of bone changes**					
Yes	154	125 (81.2%)	29 (18.8%)	0.644	-
No	36	28 (77.8%)	8 (22.2%)		
**Calcium level above 2.75 mmol/L**					
Yes	28	25 (89.3%)	3 (10.7%)	0.305	-
No	168	132 (78.6%)	36 (21.4%)		
**Creatinine above 1.9 mg/dL**					
Yes	44	28 (63.6%)	16 (36.4%)	0.005	-
No	153	129 (84.3%)	24 (15.7%)		
**ECOG**					**φc**
ECOG 0	16	14 (87.5%)	2 (12.5%)	0.866	
ECOG 1	84	68 (81.0%)	16 (19.0%)		
ECOG 2	71	54 (76.1%)	17 (23.9%)		0.090
ECOG 3	18	14 (77.8%)	4 (22.2%)		
ECOG 4	8	7 (87.5%)	1 (12.5%)		
**Physical fitness, according to Katz**					**φc**
Fully independent	158	130 (82.3%)	28 (17.7%)	0.162	0.139
Moderate impairment	23	15 (65.2%)	8 (34.8%)		
Completely dependent	11	9 (81.8%)	2 (18.2%)		
**Frailty score (IMWG)**					**rφ**
Intermediately fit	43	34 (85.0%)	9 (15.0%)	1.000	−0.010
Frailty	154	123 (79.8%)	31 (20.8%)		

Abbreviations: ECOG—Eastern Cooperative Oncology Group; ISS—International Staging System. * Cramer’s V correlation (φc); Phi correlation (rφ); IMWG—International Myeloma Working Group.

**Table 3 cancers-15-03469-t003:** Types of first-line treatments and the number of deaths during the 1st line treatment.

Name of Schema	Number (%) of Cases during First Line		Number (%) of Deaths during First Line/with Drug Modification
	Treatment with Schema without Drug Modification	Treatment with Schema with Drug Modification	
**3-Drug Schema**	**157 (79.7%)**		**12 (7.6%)/4**
CTD (cyclophosphamide, thalidomide, dexamethasone)	42 (21.3%)	12 (6.1%)	3 (5.6%)/2
VMP (bortezomib, melphalan, prednisolone)	38 (19.3%)	1 (0.5%)	6 (15.4%)/0
MPT (melphalan, prednisolone, thalidomide)	25 (12.7%)	6 (3.0%)	0/0
VTD (bortezomib, thalidomide, dexamethasone)	9 (4.6%)	13 (6.6%)	3 (13.7%)/2
VCD (bortezomib, cyclophosphamide, dexamethasone) VRD (bortezomib, lenalidomide, dexamethasone)	7 (3.6%) 1 (0.5%)	3 (1.5%) 0	0/0 0/0
**2-Drug Schema**	**40 (20.3%)**		**2 (5.0%)/1**
VD (bortezomib, dexamethasone)	5 (2.5%)	6 (3.0%)	0/0
TD (thalidomide, dexamethasone)	6 (3.0%)	3 (1.5%)	0/0
MP (melphalan, prednisolone)	4 (2.0%)	6 (3.0%)	1 (1.0%)/1
CD (cyclophosphamide, dexamethasone)	5 (2.5%)	4 (2.0%)	1 (1.1%)/0
RD (lenalidomide, dexamethasone)	1 (0.5%)	0	0/0

**Table 4 cancers-15-03469-t004:** Responses according to the 2- or 3-drug regimen.

First Line Treatment		Number (%) of Patients Responding to First-Line Treatment				
		Complete Remission (CR)	Very Good Partial Response (VGPR)	Partial Response (PR)	Disease Stabilization (SD)	Disease Progression
Regimen	2-drug	3 (7.5%)	1 (2.5%)	20 (50.0%)	14 (35.0%)	2 (5.0%)
	3-drug	15 (9.6%)	14 (8.9%)	90 (57.3%)	22 (14.0%)	16 (10.2%)
Total number		18 (9.1%)	15 (7.6%)	110 (55.9%)	36 (18.3%)	18 (9.1%)

**Table 5 cancers-15-03469-t005:** Relationship of responses to treatment to ECOG performance status, Katz scale, and the most frequently reported comorbidities.

Variable	Level	Number of Patients *n* (%)	Response to the Treatment Used *n* (%)					Independence Assessment *p*-Value
			Complete Remission (CR)	Very Good Partial Response (VGPR)	Partial Response (PR)	Disease Stabilization (SD)	Disease Progression (PD)	
**ECOG scale** ***n* = 197 (100%)**	0	16	2 (12.5%)	2 (12.5%)	8 (50.0%)	1 (6.2%)	3 (18.8%)	0.642
	1	84	11 (13.1%)	7 (8.3%)	43 (51.2%)	16 (19.0%)	7 (8.3%)	
	2	71	5 (7.0%)	5 (7.0%)	43 (60.6%)	14 (19.7%)	4 (5.6%)	
	3	18	0 (0%)	1 (5.6%)	12 (66.7%)	3 (16.7%)	2 (11.1%)	
	4	8	0 (0%)	0 (0%)	4 (50.0%)	2 (25.0%)	2 (25.0%)	
**Performance status, according to ADL (Katz)** ***n* = 192 (97.5%)**	Fully independent	158	16 (10.1%)	15 (9.5%)	89 (56.3%)	26 (16.5%)	12 (7.6%)	0.097
	Moderately impairment	23	2 (8.7%)	0 (0%)	13 (56.5%)	7 (30.4%)	1 (4.3%)	
	Completely dependent	11	0 (0%)	0 (0%)	5 (45.5%)	2 (18.2%)	4 (36.4%)	
**Number of comorbidities** ***n* = 197 (100%)**	Four and more	53	1 (1.9%)	3 (5.7%)	31 (58.5%)	11 (20.8%)	7 (13.2%)	0.156
**Cardiovascular disease *n* = 197**	**-**	166	13 (7.8%)	13 (7.8%)	92 (55.4%)	31(18.7%)	17 (10.2%)	0.504
**Kidney disease** ***n* = 133**	**-**	48	2 (4.2%)	2 (4.2%)	27 (56.2%)	11 (22.9%)	26 (12.5%)	0.388
**Creatinine above 1.9 mg/dL** ***n* = 197**	**-**	44	3(6.8%)	2(4.5%)	19 (43.2%)	13 (29.5%)	7 (15.9%)	0.058
**Diabetes** ***n* = 196**	**-**	40	3 (7.5%)	13 (8.3%|)	21 (52.5%)	9 (22.5%)	5 (12.5%)	0.753
**Cerebral circulation disorders** ***n* = 137**	**-**	18	0 (0%)	1 (5.6%)	14 (77.8%)	0 (0%)	3 (16.7%)	0.018

## Data Availability

The data presented in this study are available on request from the corresponding author. The data are not publicly available due to privacy.

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
