# Peer review of "The Real-World Evidence on the Fragility and Its Impact on the Choice of Treatment Regimen in Newly Diagnosed Patients with Multiple Myeloma over 75 Years of Age"

_cancers, 2023, doi:10.3390/cancers15133469_

Round 1
Reviewer 1 Report
An interesting study to questions the role of fragility scales in myeloma therapy. Interesting concept. Main issues are:
1. only one patient received RVD which would be considered standard of care in many developed countries - presumably this is due to availability in Poland but is a weakness of the study. Authors should comment on this.
2. The authors do NOT perform a frailty score such as IMWG but rather comment on ECOG and then monitor patients as per clinican treatment decision. This study therefore does not prove that frailty scores are not useful but that clinicians appear to be able to accurately assess frailty without a formal score.
3. This study would be much stronger scientifically if the authors were to perform one of the frailty scores and correlate with clinician choice of therapy.
Minor changes:
page 4 line 154 - change to gender neutral pronoun (themselves vs himself)
Page 4 line 194 - alter to 'one third of patients were over the age of 80'
Page 11 Discussion - percentages should have a full stop not comma. eg. line 362 with 3 drugs (4,6 %) and 8 were in the frailty group
Author Response
Dear Reviewer,
thank you for all your valuable tips and comments.
All suggestions have been taken into account and corrected as recommended.
- Thank for your valuable observation that only one patient received VRD. Indeed this is true because VRD was not reimbursed at the time on the study. However VTD in daily practice is more challenging to administer compared to VRD in the elderly population. Therefore the lake of VRD should not be considered as major limitation of our study. To clarify this issue we added in the discussion the following sentence: Of note VRD was used only in one patient since this schema was not reimbursed at the time of the study.
-
Thank you for this comment, which is partially correct, since indeed the primary physicians did not asses the frailty score. However the authors, who collected and interpreted the data performed such analysis based on the information received from primary physicians. To clarify this we added in the methodology section two sentences: The frailty scores were assessed centrally and was not disclosed to the local centers. Primary physicians did not perform the frailty scores assessment.
-
Once again thank you for this comment. We addressed this important issue in Table 3. and Table 7. supplementary, in which we showed that there is no correlation between the assessed frailty scores and primary physicians choice. However we added one more statistical analysis of frailty score (IMGW) and physicians choice of therapy. We added in Table 3. one more column with a results of correlation test between frailty scores and physicians choice.
-
page 4 line 154 - change to gender neutral pronoun (themselves vs himself) was corrected
Page 4 line 194 - alter to 'one third of patients were over the age of 80' was corrected
Page 11 Discussion - percentages should have a full stop not comma. eg. line 362 with 3 drugs (4,6 %) and 8 were in the frailty group was corrected
Best Regards Agata

Reviewer 2 Report
Authors provide an interesting study highlighting that in real world fragilty score does not have a significant impact on patient response to two or three drug combinations. This is fitting for this special issue and addresses an important area with unmet needs. Although authors do give details on the specifics of drug combinations in Table 2 alongside number of deaths, downstream analyses do not compare responses to each specific drug combination within each fragility group. This would be useful to add in and add more weight to conclusions. There are a few other weaknesses, however, the authors do acknowledge these themselves e.g. the lack of cytogenetic data and most patients falling into categories ECOG 0-2. Addition of a PFS curve should be added with groups stratified by fragility & drug combination, I understand the follow-up is relatively short at 12-months but even addition to the supplementary files would be useful.
Again, authors have shown in table 5 the response to treatment as a whole in patients grouped by different fragility measures but should add similar analyses for that includes further subgrouping by 2 or 3 drug combination received. Similarly, section 3.7 analysis of deaths should be related back to the fragility assessment.
Typographical/grammatical errors:
Throughout manuscript and supplementary the use of points (.) and commas (,) as decimal separators vary, please choose one of these and keep it consistent throughout.
Paragraph beginning line 200 – references to table 2 (line 201) and table 3 (line 203) are mixed up.
Line 68/69 – ‘Three-drugs regimens received 34 patients from the intermediate-fit group (79.0%), and 123 (79.9%) from the frail 69 group’ – please re-word doesn’t make sense.
Table 1 – please check number of cases for each characteristic, for ‘bone disease Yes/No’ section the numbers add up to 190 not 197. If this is because information for all 197 patients is not available, please state this or add in number of cases for each characteristic in separate column.
I appreciate that abbreviations for drug regimens are explained in table 2 but perhaps add this into main text (section 3.3) since this is not a multiple myeloma special issue.
Table 2 – VRD typo – ‘lenalidomid’
Line 251-4 ‘Every fourth person had at least four comorbidities, and 71.7% of the patients in this group received a three-drug treatment The percentages of patients with a lower number of comorbidities receiving 3-drug schemas were 253 similar and comparable to patients with a higher number of comorbidities Figure 1.’ - please re-word, maybe ‘25% of patients had at least four comorbidities’ . Also need brackets around figure 1 and addition of punctuation.
Author Response
Dear Reviewer,
Thank you very much for your important comment. However we intentionally do not provided data regarding the response to the treatment since it was not a major goal of our study. We only wanted to check if there is a correlation between the frailty scores and primary physicians scores available at the time of the study. Moreover the responses will change depending on the available treatment options and might be addressed in the future together with data collected from real-life practice, currently used regimens such as Dara-VMP, Dara-Rd, CART and bispecific antibodies.
In our study we focused on the early period (1 year) of the treatment to find out if the frailty score predicted early toxicity and mortality. We didn’t collect data to assess PFS in this group of patients in the longer follow-up.
The stratification according to the next variable proposed by the reviewer will "blur" the numbers in the subgroups. Of course we can multiply subsequent tables, but many fields will remain empty and in general there will be few conclusive observations.
Typographical/grammatical errors:
Throughout manuscript and supplementary the use of points (.) and commas (,) as decimal separators vary, please choose one of these and keep it consistent throughout. was corrected
AND:
Paragraph beginning line 200 – references to table 2 (line 201) and table 3 (line 203) are mixed up. was corrected
Line 68/69 – ‘Three-drugs regimens received 34 patients from the intermediate-fit group (79.0%), and 123 (79.9%) from the frail 69 group’ – please re-word doesn’t make sense. was corrected
Table 1 – please check number of cases for each characteristic, for ‘bone disease Yes/No’ section the numbers add up to 190 not 197. If this is because information for all 197 patients is not available, please state this or add in number of cases for each characteristic in separate column. was corrected
I appreciate that abbreviations for drug regimens are explained in table 2 but perhaps add this into main text (section 3.3) since this is not a multiple myeloma special issue. was corrected
Table 2 – VRD typo – ‘lenalidomid’ was corrected
Line 251-4 ‘Every fourth person had at least four comorbidities, and 71.7% of the patients in this group received a three-drug treatment The percentages of patients with a lower number of comorbidities receiving 3-drug schemas were 253 similar and comparable to patients with a higher number of comorbidities Figure 1.’ - please re-word, maybe ‘25% of patients had at least four comorbidities’ . Also need brackets around figure 1 and addition of punctuation. was corrected
Thank you for your detailed assessment of the proposed changes.
We incorporated all your other suggestions into the manuscript.
Best Regards Agata

Reviewer 3 Report
The paper, is not original and, above all, despite it is a prospective study it appears to be obsolete considering that it refers to elderly patients treated between 2018 and 2019 with regimens as CTD, VPM, MPT or TD and VD that belong to a previous era of MM treatment. All frailty scores that the Authors cited in the introduction have been developed in a population receiving “old” therapy but I wonder whether it could perform in the same way using new regimens as Dara-VMP or Dara-Rd. The Authors stated in the Discussion that older patients benefit from the three-drug regimens but we know that all these patients benefit from quadruplets, representing the current standard of care. The most recent subanalyses of MAIA trial presented at the last ASH meeting, showed a median PFS of 54.3 months in patients aged ≥ 75 years receiving D-Rd (vs 31 months for Rd) (Moreau et al, poster 3245), being median OS of 73.5 years in the same group (vs 54.8 for Rd) (Kumar et al, poster 4559). Moreover, after a 36.4-month median follow-up, median PFS was NR vs 30.4 months in frail patients (HR, 0.62; 95% CI, 0.45–0.85; P = 0.003) (Facon et al, Leukemia 2022). I think two drug regimens can not currently be an option for patients who are not eligible for ASCT. Moreover, future frailty scores will have to be based on more objective data and certainly not on “physician feeling”, considering the potential severe toxicities of new immunotherapies as bispecific antibodies and CAR T cells. There are not references 25 and 26.
No comments
Author Response
Dear Reviewer,
Thank you very much for your valuable comment. Indeed the used regiments in our study belong to the previous era of the multiple myeloma treatment since in Poland the introduction of the new treatment was delayed. Nevertheless we would like to stress that main goal of our study was to find out if there is a correlation between the physicians choice and frailty score. We have showed in our paper that such correlation does not exist since even in the previous era of multiple myeloma treatment. New treatment options for elderly patients such as Dara-Rd, Dara-VMP and bispecific antibodies and CAR T will may change the utility of the frailty scores in the daily practice but this needs to be shown in the future studies. We completely agree with the reviewer that future frailty scores will have to be based on more objective data and not on “physician feeling”. We modified the discussion according to the reviewer’s comment: However new era in the multiple myeloma treatment may require to re-address the utility of the frailty scores. Future frailty scores should be based on more objective data and not on “physician feeling”, especially in view of the potential severe toxicities of new treatment options (immunotherapy and CAR T cells).
Best Regards Agata

Reviewer 4 Report
The manuscript describes important novel data in the studied area and worth to be published with some improvements in the presentation style and language. Discussion section shall be expanded via additional perspectives regarding previous papers in the Myeloma-prognostication field (such as; Evaluation of Prognostic Significance of the International Staging System According to Glomerular Filtration Rate in Newly Diagnosed Multiple Myeloma Patients Eligible for Autologous Stem Cell Transplantation.
Turk J Haematol. 2021 Feb 25;38(1):33-40. doi: 10.4274/tjh.galenos.2020.2020.0115. Epub 2020 Jun 16.
may be improved
Author Response
Dear Reviewer,
thank you for your valuable comment. We added this missing citation and change the discussion accordingly: Renal impairment usually indicates more aggressive and advanced disease not always well reflected by ISS score [25].
Best Regards Agata

Round 2
Reviewer 2 Report
Authors have justified comments made.
N/A
Reviewer 3 Report
Thank for comments of Authors